# Assessment of the Effects of Physiotherapy on Back Care and Prevention of Non-Specific Low Back Pain in Children and Adolescents: A Systematic Review and Meta-Analysis

**DOI:** 10.3390/healthcare12101036

**Published:** 2024-05-16

**Authors:** José Manuel García-Moreno, Inmaculada Calvo-Muñoz, Antonia Gómez-Conesa, José Antonio López-López

**Affiliations:** 1International School of Doctoral Studies, University of Murcia, 30100 Murcia, Spain; josemanuel.garcia10@um.es; 2Department of Physiotherapy, UCAM Catholic University of Murcia, Guadalupe, 30007 Murcia, Spain; 3Research Group Research Methods and Evaluation in Social Sciences, Mare Nostrum Campus of International Excellence, University of Murcia, 30100 Murcia, Spain; agomez@um.es (A.G.-C.); josealopezlopez@um.es (J.A.L.-L.); 4Department of Basic Psychology and Methodology, University of Murcia, 30100 Murcia, Spain

**Keywords:** low back pain, non-specific low back pain, children, adolescents, physiotherapy, prevention, meta-analysis

## Abstract

Non-specific low back pain (NSLBP) in children and adolescents has increased in recent years, and the evidence of the physiotherapy interventions in back care needs to be updated. Our main goal was to quantify the effects of preventive physiotherapy interventions on improving behavior and knowledge related to back care and prevention of NSLBP in children and adolescents. Based on two previous meta-analyses, Cochrane Library, MEDLINE, PEDro, Web of Science, LILACS, IBECS, PsycINFO, and IME databases and several journals were searched. Two researchers independently extracted data and assessed the risk of bias in the studies using the RoB2 tool. Data were described according to PRISMA guidelines. A total of 24 studies (28 reports) were included. In the posttest, the behavior variable obtained an overall effect size of d_+_ = 1.48 (95%CI: 0.40 to 2.56), and the knowledge variable obtained an effect size of d_+_ = 1.41 (95%CI: 1.05 to 1.76). Physiotherapy has demonstrated beneficial impacts on behavior and knowledge concerning back care and to prevent NSLBP in children and adolescents. Interventions focusing on postural hygiene and exercise should be preferred, especially those that are shorter in number of weeks, more intense, and incorporate as many intervention hours as possible.

## 1. Introduction

Low back pain (LBP) has become a major public health concern, especially as its incidence among children and adolescents has increased in recent times [1,2]. Research suggests that approximately 39% of people aged 9 to 16 years suffer LBP at some point during this stage and that the prevalence resembles that of adults by the age of 15 years [3]. It should be noted that having LBP during childhood and adolescence increases the probability of having it in adulthood too [2,4]. The most common form of LBP is non-specific low back pain (NSLBP) [1].

The appearance of LBP in childhood and adolescence can have serious consequences, such as limitations in activities of daily living, restrictions in participation in sports activities and in the school environment, and even cases of school absenteeism. Therefore, it is essential to prioritize preventive measures against this condition [5,6].

Physiotherapists use treatments for back care and prevention of NSLBP, given that adolescents [7,8] and parents [9] do not usually have knowledge about back care. The treatments developed focus on improving knowledge about back care based on postural hygiene (theory or practice) and through therapeutic exercises to reinforce the acquired information. Empowering children and adolescents with this knowledge enables them to make lifestyle changes independently, adopting habits that promote a healthier back and overall well-being.

Another method for promoting back care consists of modifying behavior during daily activities that can have an impact on the back [5,6], such as ensuring the proper use of school backpacks [9], maintaining uniform posture changes [5], lifting weights from the floor correctly [6], and improving sitting and standing postures for prolonged periods of time [5,6]. Teaching these behavioral changes, along with promoting knowledge through postural hygiene and physical exercises associated with back care, can help solidify the concepts learned [5,6,10]. In adults, clinical trials [11] and meta-analyses [12] have shown that the combination of exercise and education has a strong preventive effect. Adapting this approach to the youth population presents an interesting possibility.

The present study aimed to quantify the effects of preventive physiotherapy interventions on improving behavior and knowledge related to back care and prevention of NSLBP in children and adolescents. Although previous systematic reviews [13] and meta-analyses addressed this question [14,15], a meta-analysis compiling all studies published to date has not yet been performed. Therefore, a comprehensive and up-to-date analysis on the effects of physiotherapy on back care in this population is required.

## 2. Material and Methods

The meta-analysis was conducted following the Preferred Reporting Items for Systematic Reviews and Meta-Analyses (PRISMA) guidelines [16]; for more information, see Appendix A. Additionally, the study was registered with PROSPERO (CRD42024510058).

### 2.1. Eligibility Criteria

The inclusion and exclusion criteria were previously established using the participants, interventions, comparators, outcomes, and study design (PICOS) strategy.

The criteria applied were: (1) participants under 18 years of age, who did not have spinal pathologies or other pathologies causing LBP; (2) the interventions had to be preventive physiotherapy, including education, therapeutic exercise, and physical activity, alone or in combination; (3) the studies had to compare at least one experimental group with a control group, and there could be several experimental groups in the same study; (4) the results should measure the knowledge and/or behavior of the participants, in a pretest and posttest evaluation using the same tool, providing sufficient statistical information to perform the analyses (sample size, mean, and SD); (5) the studies had to be controlled clinical trials, randomized and non-randomized, including published and unpublished studies of any type (journal study, doctoral thesis, conference proceedings, etc.).

### 2.2. Data Sources and Search

The research included in this study comprises data gathered from two previously conducted meta-analyses [14,15]. The search spanned from inception to May 2012 in meta-analysis 1 [14] and from May 2012 to May 2020 in meta-analysis 2 [15]. Several databases were used for data collection, including Cochrane Library, MEDLINE, PEDro, Web of Science, and IME, along with journals from the Elsevier Iberoamerican database for meta-analysis 1. For meta-analysis 2, data were collected from Cochrane Library, MEDLINE, PEDro, Web of Science, LILACS, IBECS, and PsycINFO databases, as well as from specialized journals such as BMJ and Spine. Both meta-analyses also involved a search for unpublished studies and a thorough review of the bibliographies of the included studies. For more information, see Appendix A. 

### 2.3. Study Selection

The studies were included based on the two meta-analyses previously mentioned [14,15]. In addition, two reviewers (JMGM and ICM) independently screened the studies to confirm that they met the inclusion criteria; in case of discrepancy, a third author (AGC) would decide. Cohen’s kappa was used to assess inter-rater agreement, with a result of 1.

### 2.4. Data Extraction Process

The data extracted from the two previous meta-analyses [14,15] were used. These data were extracted individually by two authors in their corresponding studies. For the present study, one author (JMGM) extracted the data from the two previous meta-analyses with the review of a second author (ICM) and, in the case of discrepancy, a third author would decide (AGC) but this was not necessary because there was total agreement. When required, additional data were requested directly from the corresponding authors.

Different variables were extracted from each study based on a previously established manual based on Lipsey’s recommendations [17]. The variables have been grouped into three different categories: substantive (treatment, context, and participant), methodological, and extrinsic variables. For more information about the coded variables, see Appendix A.

### 2.5. Assessment of Methodological Quality, Risk of Bias, and Certainty of Evidence

The Risk of Bias 2 (RoB2) tool was used to analyze the risk of bias [18]. This tool allows a clinical trial to be judged as “low risk of bias”, “some concerns”, and “high risk of bias”. Each study was analyzed in each of its domains and the final assessment was the worst of the domains, or if a study had several domains where “some concerns” where found it was rated as “high risk of bias”, as recommended by the authors of the tool. 

The GRADE system was applied for each of the variables studied in this meta-analysis.

Two authors performed the analysis of risk of bias and certainty of evidence separately (JMGM and ICM) and, in case of discrepancy, a third author would decide (AGC) but this was not necessary since the agreement was total.

### 2.6. Statistical Analysis

A random-effects model was applied [19] with the correction proposed by Hartung [20]. A forest plot, including 95% confidence intervals, was created to represent numerically and graphically the individual effects of each study, as well as to display the average effect size. The prediction interval was also added. To calculate the effect size, a standardized mean difference “*d*” was used for quantitative variables [21] (Equations (6) and (7) were used to quantify the standardized mean change in each group and its sampling variance, and then the d index was calculated as the difference using Equations (10) and (11) of ref. [21]).

Four included studies were three-arm trials with two experimental groups [22,23,24,25]. For these, the effect size was calculated separately for each experimental group in relation to the control group. While this violates the independence assumption required when applying standard meta-analytic techniques (as each participant allocated to control groups in those studies contributed to two effect sizes), we note that the extent of multiplicity in our database—and consequently the potential for statistical dependency—is minimal given the reduced number of studies affected and the small size of the groups involved compared to the rest of the included studies (see Table 1). Therefore, standard meta-analytic models were preferred for this review over other methodological options that either involve loss of information or additional statistical complexity [26].

To assess heterogeneity, the I^2^ index was used and interpreted in combination with 95% prediction intervals. 

For the analysis of moderator variables, weighted ANOVA was used for qualitative moderator variables and meta-regression for continuous moderator variables. Both analyses were corrected as proposed by Knapp and Hartung [27]. For the analysis of publication bias, Egger’s test and a funnel plot were applied for all variables. All statistical analyses were performed using R [28] in conjunction with the metafor package [29].

**Table 1 healthcare-12-01036-t001:** Characteristics of the studies.

Study	Country	Design	Participants	Intervention	Intervention Description	Intervention Duration	Main Results
Spence et al., 1984 [22]	United States	RCT	E1 *n* = 25 Age (SD) = NA % Male = NA	Postural hygiene	Knowledge acquisition, theoretical (demonstration lecture and group discussion) 10-min interactive teaching session about safe lifting techniques	1 week, 0.16 h per week, 0.16 total hours	Both groups E1 and E2 obtained statistically significant results in the knowledge in the posttest in comparison to control group. Control group obtained non-statistically significant results in the behavior in the posttest in comparison to E1 and E2
			E2 *n* = 26 Age (SD) = NA % Male = NA	Postural hygiene	Knowledge acquisition, theoretical (guided discovery) 15 min of guided self-discovery session	1 week, 0.25 h per week, 0.25 total hours	
			C *n* = 25 Age (SD) = NA % Male = NA	No intervention	No intervention		
Cardon et al., 2000 [30]	Spain	RCT	E *n* = 42 Age (SD) = 9.93 % Male = 38.1	Postural hygiene + exercise	Knowledge acquisition + postural habits training (picking up, carrying, and handling of weights, working positions and activities of daily living and resting positions), theoretical (guided discovery) and practical + stretching + strengthening + pelvic tilt + relaxing Based on guided discovery and active hands-on method, good posture in activities of daily living and weight lifting, anatomy and pathology of the spine, pelvic tilt exercises, stretching and strengthening of abdominals and squats and relaxation exercises were taught. For parents and teachers, information on back care was also provided	6 weeks, 1 h per week, 6 total hours	The experimental group obtained a statistically significantly higher score than the control group for the knowledge in the posttest
			C *n* = 26 Age (SD) = 11.1 % Male = 52.8	No intervention	No intervention		
Gómez and Méndez, 2000a [31]	Spain	RCT	E *n* = 33 Age (SD) = 11 % Male = 45	Postural hygiene	Knowledge acquisition, theoretical (demonstration lecture and group discussion) Information on anatomy and biomechanics of the spine, respiratory mechanism and how to avoid spinal overload	8 weeks, 0.5 h per week, 4 h in total	The experimental group obtained statistically significantly higher scores than the control group for the knowledge and not statistically significantly higher scores for behavior in the posttest
			C *n* = 34 Age (SD) = NA % Male = 50	Regular academic activities on related topics	NA	8 weeks, 0.5 h per week, 4 h in total	
Gómez and Méndez, 2000b [23]	Spain	RCT	E1 *n* = 33 Age (SD) = 11 % Male = 45.45	Postural hygiene	Knowledge acquisition + postural habits training (picking up, carrying, and handling of weights, working positions and activities of daily living and resting positions), theoretical A physiotherapist provided the children with information on healthy postural habits. In addition, the children’s parents were provided with information on postural hygiene habits and training in observation and registration of postural habits	12 weeks, 0.16 h per week, 4 total hours	Both E1 and E2 obtained non-statistically significant results in comparison to control group in the behavior in the posttest
			E2 *n* = 34 Age (SD) = 11 % Male = 50	Postural hygiene	Knowledge acquisition + postural habits training (picking up, carrying, and handling of weights, working positions and activities of daily living and resting positions), theoretical The teacher-tutor provided the children ergonomic advice. In addition, parents were provided with information on postural hygiene habits and training in observation and recording of postural habits	12 weeks, 0.16 h per week, 4 total hours	
			C *n* = 32 Age (SD) = NA % Male = 43.75	Postural hygiene	Parents were provided with information on postural hygiene habits and training in observation and recording of postural habits	NA	
Cardon et al., 2001 [24]	Belgium	RCT	E1 *n* = 38 Age (SD) = 10.9 (0.6) % Male = 42.1	Postural hygiene	Knowledge acquisition + postural habits training (picking up, carrying, and handling of weights, working positions and activities of daily living and resting positions), theoretical (guided discovery) and practical Based on guided discovery and active hands-on method, good posture in activities of daily living and how to carry backpack were taught. Teachers and parents also received this information and teachers were invited to participate in the sessions. Each participant obtained a back care manual that included: keeping the natural curvatures of the back, encouragement to be active, correcting working tables, how to lift weights, and how to carry books in a bag	15 weeks	Both E1 and E2 obtained non-statistically significant results in comparison to control group in the knowledge and statistically significant results in the behavior in the posttest
			E2 *n* = 48 Age (SD) = 11.1 (0.7) % Male = 43.75	Postural hygiene	Knowledge acquisition + postural habits training (picking up, carrying, and handling of weights, working positions and activities of daily living and resting positions), theoretical (guided discovery) and practical Based on guided discovery and active hands-on method, good posture in activities of daily living and how to carry backpacks were taught	6 weeks, 1 h per week, 6 total hours	
			C *n* = 34 Age (SD) = 11 (0.6) % Male = 50	No intervention	No intervention		
Méndez and Gómez, 2001 [32]	Spain	RCT	E *n* = 35 Age (SD) = 9 % Male = 54.28	Postural hygiene + exercise	Knowledge acquisition + postural habits training (picking up, carrying, and handling of weights, working positions and activities of daily living and resting positions), theoretical (guided discovery, demonstration class, and group discussion) and practical + stretching, strengthening, pelvic tilt, breathing, and postural correction Parents were given information about postural hygiene in general. Children were taught in class about the importance of correct posture, prevention of back pain, and the respiratory system. Training in everyday situations, such as sitting, writing, eating, watching TV, etc. Exercises to strength the abdominal and dorsal muscles and balancing the pelvis. Encouragement to do the exercises at home in an everyday routine	8 weeks, 2.375 h per week, 19 total hours	The experimental group obtained statistically significant results in comparison to control group in the knowledge and behavior in the posttest
			C *n* = 35 Age (SD) = 9 % Male = 51.43	Postural hygiene + exercise	Different academic activities with related topics: disease prevention, healthy habits, spine, respiratory system, differences between subjects in terms of body development, physical exercise, muscle training, and postural biomechanics in human behavior	NA	
Cardon et al., 2002a [33]	Belgium	RCT	E *n* = 347 Age (SD) = 10 (0.6) % Male = 47.6	Postural hygiene	Knowledge acquisition + postural habits training (picking up, carrying, and handling of weights, working positions and activities of daily living and resting positions), theoretical (guided discovery) and practical Based on guided discovery and active hands-on method, anatomy and pathology of the back and basic principles of correct posture in daily activities were taught. For teachers and parents, an information session was organized based on back care principles to enhance learning at home and in the classroom. The teachers received a manual with the information and extra exercises	6 weeks, 1 h per week, 6 total hours	The experimental group obtained non-statistically significant results in comparison to control group in the behavior and statistically significant results in the knowledge in the posttest
			C *n* = 359 Age (SD) = 10.1 (0.7) % Male = 65.7	No intervention	No intervention		
Cardon et al., 2002b [34]	Belgium	RCT	E *n* = 198 Age (SD) = 9.8 % Male = 47.5	Postural hygiene	Knowledge acquisition + postural habits training (picking up, carrying, and handling of weights, working positions and activities of daily living and resting positions), theoretical (guided discovery) and practical Based on guided discovery and active hands-on method, anatomy and pathology of the back and basic principles of correct posture in daily activities were taught. For teachers and parents, an information session was organized based on back care principles to enhance learning at home and in the classroom. The teachers received a manual with the information and extra exercises	6 weeks, 1 h per week, 6 total hours	The experimental group obtained statistically significant results in comparison to control group in the behavior in the posttest
			C *n* = 165 Age (SD) = 10.3 % Male = 46.7	No intervention	No intervention		
Geldhof et al., 2006 [35]	Belgium	RCT	E *n* = 214 Age (SD) = 11.3 (0.8) % Male = 48.18	Postural hygiene	Knowledge acquisition + postural habits training (picking up, carrying, and handling of weights, working positions and activities of daily living and resting positions) and stimulation of dynamic postures in class, theoretical (guided discovery) and practical Back education based on anatomy and pathology of the back, principles of biomechanical postural behavior and skills in accordance with good body mechanics. Teachers completed the integration of the back postural principles learned weekly with the use of imagery. In addition, the physical education teacher taught the improvement of dynamic sitting (active and variable sitting), the interruption of prolonged static sitting, and the activating approach (changes in class organization, promotion of postural behavior, and activating methodology). The teacher’s intervention lasted 2 years	96 weeks	The experimental group obtained statistically significant higher scores than the control group for the behavior and knowledge in the posttest
			C *n* = 184 Age (SD) = 11.4 (0.8) % Male = 47.67	No intervention	No intervention		
Cardon et al., 2007 [36]	Belgium	RCT	E *n* = 205 Age (SD) = NA % Male = NA	Postural hygiene + physical activity	Knowledge acquisition + postural habits training (picking up, carrying, and handling of weights, working positions and activities of daily living and resting positions), dynamic postural stimulation in the school classroom, theoretical (guided discovery) and practical + sport, games, and active recess Based on guided discovery and active hands-on method, children were taught basic anatomy and pathology of the back and basic principles of biomechanically favorable postures during daily activities. Additionally, class teachers were given guidelines to increase postural dynamics in the classrooms. A promotion of physical activity inside and outside school and to develop an active lifestyle. Extra-curricular sports sessions and games were implemented weekly, and children received materials to be used during recess and lunch breaks	96 weeks	The experimental group obtained non-statistically significantly higher scores than the control group for the behavior and knowledge in the posttest
			C *n* = 184 Age (SD) = NA % Male = NA	No intervention	No intervention		
Martínez, 2007 [37]	Spain	RCT	E *n* = 314 Age (SD) = 9.46 (1.26) % Male = NA	Postural hygiene	Knowledge acquisition + postural habits training (picking up, carrying, and handling of weights, working positions and activities of daily living and resting positions), theoretical (guided discovery and demonstrative talk) and practical Ergonomics and postural hygiene based on activities of daily living such as lying down and getting out of bed, how to lie in bed, getting in and out of a car, washing hands, sitting on a high stool, using pillows to rest, carrying weight, lifting weight, lifting small objects, how to sit in a chair and how to get up, reading in bed, sitting to study, working on the computer, backpack use (preparation and carrying), carrying small cabinets, using ladders to pick up objects at height, watching TV, sitting to eat, working while standing, ironing while standing, and in the last session a general review of all of the above	5 weeks, 1 h per week, 5 total hours	The experimental group obtained statistically significantly higher scores than the control group for the knowledge in the posttest
			C *n* = 265 Age (SD) = 9.66 (1.19) % Male = NA	No intervention	No intervention		
Cardoso et al., 2009 [38]	Brazil	RCT	E *n* = 269 Age (SD) = NA % Male = NA	Postural hygiene	Knowledge acquisition + postural habits training (picking up, carrying, and handling of weights, working positions and activities of daily living and resting positions), theoretical (demonstrative talk) and practical Based on demonstrative talk, children were taught spinal care principles and how to incorporate this knowledge into daily life	2 weeks, 1.33 h per week, 2.66 total hours	The experimental group obtained statistically significantly higher scores than the control group for the knowledge in the posttest
			C *n* = 250 Age (SD) = NA % Male = NA	No intervention	No intervention		
Vidal, 2009 [39]	Spain	RCT	E *n* = 63 Age (SD) = 10.7 % Male = NA	Postural hygiene + exercise	Postural habits training (picking up, transport of weights), theoretical and practical + exercise (breathing, postural correction, balance, and relaxation) Anatomy, biomechanics, risk factors for injury, spinal care principles, respiratory mechanism, postural hygiene, and behavioral intervention	4 weeks, 1.75 h per week, 7 total hours	The experimental group obtained non-statistically significantly higher scores than the control group for the knowledge in the posttest
			C *n* = 74 Age (SD) = 10.7 % Male = NA	No intervention	No intervention		
Kovacs et al., 2011 [40]	Spain	RCT	E *n* = 320 Age (SD) = 8 % Male = 55.5	Postural hygiene	Knowledge acquisition (theoretical) Use of the *Comic Book of the Back* (prevention and management of LBP)	NA	The experimental group obtained non-statistically significantly higher scores than the control group for the knowledge in the posttest
			C *n* = 254 Age (SD) = 8 % Male = 49.8	No intervention	No intervention		
Park and Kim, 2011 [25]			E1 *n* = 28 Age (SD) = 11.96 (0.96) % Male = 50	Postural hygiene	Knowledge acquisition (theoretical) Web-based spinal health education program (anatomy, functions of the spine, spinal care principles, stretching and strengthening exercises, backpack use)	4 weeks, 0.5 h per week, 2 total hours	Both E1 and E2 obtained statistically significant results in comparison to control group in the knowledge in the posttest and non-statistically significant results in comparison to control group in the behavior in the posttest
			E2 *n* = 29 Age (SD) = 12 (0) % Male = 51.72	Postural hygiene	Knowledge acquisition (theoretical) Face-to-face spinal health education program (anatomy, functions of the spine, spinal care principles, stretching and strengthening exercises, backpack use)	4 weeks, 0.5 h per week, 2 total hours	
			C *n* = 31 Age (SD) = 11.94 (0.18) % Male = 61.29	No intervention	No intervention		
Hashemi et al., 2012 [41]	Iran	RCT	E *n* = 203 Age (SD) = NA % Male = 50.2	Postural hygiene	Knowledge acquisition + postural habits training Anatomy and structure of spine, ergonomics about backpacks + sitting posture and lying, body posture while lifting, pushing, and pulling (theoretical and practical)	4 total hours	The experimental group obtained statistically significantly higher scores than the control group for the behavior and knowledge in the posttest
			C *n* = 201 Age (SD) = NA % Male = 48.3	No intervention	No intervention		
Gallardo et al., 2013 [42]	Spain	RCT	E *n* = 271 Age (SD) = 8.8 (0.69) % Male = 43.2	Postural hygiene	Knowledge acquisition + postural habits training Requirements of a school backpack: size, capacity, and for its adequate transport. (1) How to carry a backpack on your back properly: adjustment, placement, and position of the back to walk, (2) how to properly transport a wheeled backpack: handle adjustment, upper limb placement and shape to drag or push, and (3) how to properly select and order school material in a backpack	3 weeks, 0.75 h per week, 18 total hours	The experimental group obtained statistically significantly higher scores than the control group for the behavior in the posttest
			C *n* = 87 Age (SD) = 8.6 (0.72) % Male = 57	No intervention	No intervention		
Ritter and de Souza, 2015 [43]	Brazil	Non-RCT	E *n* = 26 Age (SD) = 14 (0.93) % Male = 31.35	Postural hygiene + exercise + physical activity	Knowledge acquisition + postural habits training (theoretical and practical) + stretching + games Evolution of humans and the spine, the emergence of spinal curves in humans: from birth to adulthood, role of spinal curves and spine structures + sitting, standing, rising from a chair, sitting to write, picking up objects from the floor and carrying schoolbags + stretching + recreational and associative activity	10 weeks, 1.66 h per week, 16.66 total hours	The experimental group obtained statistically significantly higher scores than the control group for the behavior in the posttest
			C *n* = 23 Age (SD) = 15.38 (0.97) % Male = 44.83	No intervention	No intervention		
Sellschop et al., 2015 [44]	South Africa	RCT	E *n* = 61 Age (SD) = 13.4 (0.7) % Male = 61	Postural hygiene	Knowledge acquisition + postural habits training (theoretical and practical) Carrying a school bag correctly, co-operative group work with problem-solving tasks related to poor postural habits and computer work	1 week, 0.75 h per week, 0.75 total hours	The experimental group did not obtain statistically significantly higher scores than the control group for the behavior in the posttest
			C *n* = 66 Age (SD) = 13.4 (0.5) % Male = 59	No intervention	No intervention		
Brzek and Plinta, 2016 [45]	Poland	Non-RCT	E *n* = 144 Age (SD) = 7.6 (0.64) % Male = 56.9	Postural hygiene	Knowledge acquisition + postural habits training (theoretical and practical) Anatomy and function of the spine, causes of postural disorders, how bad postures may affect adults, ergonomics in daily activities, weight of school bag + good positions to carry the school bag	NA	The experimental group obtained statistically significantly higher scores than the control group for the behavior in the posttest
			C *n* = 222 Age (SD) = 7.72 (0.73) % Male = 55.86	No intervention	No intervention		
Dullien et al., 2018 [46]	Germany	RCT	E *n* = 90 Age (SD) = 10.59 (0.43) % Male = 48.18	Postural hygiene + exercise	Knowledge acquisition + postural habits training (theoretical) + stretching + strengthening Anatomy + good and bad posture while sitting, healthy backpack habits, healthy lifting and carrying, back-friendly sports and nutrition + stretching + strengthening of back (hip lifts and ball exercises) and abdominal muscles (plank, crunch and ball exercises) (additionally, the use of posters on posture awareness, strengthening exercises, and stretching exercises)	12 weeks, 0.31 h per week, 3.72 total hours	Both groups obtained better results in their posttest and 6-month follow-up measurements. The addition of manipulation to treatment resulted in non-significant improvements in LBP intensity at posttreatment evaluation and at 6-month follow-up in adolescents
			C *n* = 86 Age (SD) = 10.52 (0.42) % Male = 48.18	No intervention	No intervention		
Sellschop et al., 2018 [47]	South Africa	RCT	E *n* = 61 Age (SD) = 13.4 (0.7) % Male = 61	Postural hygiene + exercise	Knowledge acquisition + postural habits training (theoretical) + stretching Posture, backpack weight + workstation set-up + neck, shoulder, and lower back stretches	1 week, 0.75 h per week, 0.75 total hours	The experimental group obtained statistically significantly higher scores than the control group for the behavior in the posttest
			C *n* = 66 Age (SD) = 13.4 (0.5) % Male = 59	No intervention	No intervention		
Miñana-Signes et al., 2019 [48]	Spain	Non-RCT	E *n* = 16 Age (SD) = 15.5 (1.6) % Male = 30.1	Postural hygiene + exercise + physical activity	Knowledge acquisition + postural habits training (theoretical and practical) + stretching + strengthening + pelvis lift + postural correction + relaxation + sports + games Anatomy, functions of the back, most common pathologies + correct and incorrect postural habits, sitting, lifting objects, transporting objects, sleeping, writing, sweeping, brushing teeth, using a mobile phone, carrying a backpack + stretching of hamstrings, lumbar quadrate, paravertebral, latissimus dorsi, iliopsoas + strengthening the trunk musculature (abdominal and lumbar isometric) + pelvis lift + postural correction + relaxation (Jacobson) + football and floorball + racing games	2 weeks, 2.62 h per week, 5.25 total hours	The experimental group obtained statistically significantly higher scores than the control group for the knowledge and behavior in the posttest
			C *n* = 16 Age (SD) = 15.3 (1.8) % Male = 32.6	Treatment as usual	Usual physical education classes		
Akbari-Chehrehbargh et al., 2020 [49]	Iran	RCT	E *n* = 52 Age (SD) = 11 (1) % Male = 0	Postural hygiene + exercise	Knowledge acquisition + postural habits training (theorical and practical) + stretching + strengthening Spine anatomy, natural curves in the spine and during daily activities and back care knowledge + skills training activities based on experience and practical demonstrations, backpack wearing, carrying objects, proper sitting and standing postures + back strengthening + stretching	6 weeks, 1 h per week, 6 total hours	The experimental group obtained statistically significantly higher scores than the control group for behavior and knowledge in the posttest
			C *n* = 52 Age (SD) = 11 (1) % Male = 0	No intervention	No intervention		

RCT: randomized controlled trial, NA: not available, E: experimental, C: control, SD: standard deviation.

## 3. Results

### 3.1. Study Selection

After searching the articles following the previously mentioned search strategy, a total of 4107 articles were found. After elimination of duplicates, 4058 articles were chosen for examination. After review of the titles and abstracts, 121 articles were selected to be thoroughly analyzed to determine whether they met the inclusion criteria. Finally, 24 studies were included resulting in 28 reports. The flow chart (Figure 1) provides a detailed description of the article selection process.

### 3.2. Study Characteristics

The 24 studies were published between 1984 and 2020 [22,23,24,25,30,31,32,33,34,35,36,37,38,39,40,41,42,43,44,45,46,47,48,49]. All studies were RCTs, except for three that were quasi-experimental controlled studies [43,45,48]. All studies had an experimental group and a control group, except for four studies that had a control group and two experimental groups [22,23,24,25], therefore, an analysis of each experimental group with the control group was performed independently. While we recognize the potential for statistical dependencies, these are expected to have little impact on the results given the small number of studies with multiple comparisons [26].

All studies were journal articles except for two doctoral theses [37,38]. The studies were carried out in Spain [23,31,32,37,39,40,42,48], the United States [22], Belgium [24,30,33,34,35,36], Brazil [38,43], South Korea [25], Iran [41,49], South Africa [44,47], and Germany [46]. Regarding the first author, in most of the articles it was a physiotherapist [22,23,24,30,31,33,34,35,36,37,38,42,44,45,47], and rarely a medic [40], nurse [25], physical education teacher [39,43,48], psychologist [32], or not specified [41,46,49]. All studies were conducted at a school and the participants were students. For more information, see Table 1.

#### 3.2.1. Sample

The number of participants in the included studies varied, ranging from 32 [48] to 706 [33]. In the initial assessment (pretest), the study included 5996 participants in total: 3241 in the experimental groups and 2755 in the control groups. Following the posttest, the total number of participants decreased to 5869, with 3155 in the experimental groups and 2714 in the control groups. The age mean ranged from 8 years [40] to 15.38 years [43]. The age groups were mostly children, a few studies included only adolescents [43,47], and one study included both children and adolescents [38]. The gender of the participants spanned from 0% [49] to 61% male [47], and in some studies it was not specified [22,36,37,39]. 

#### 3.2.2. Intervention

Regarding the intervention, the experimental groups performed postural hygiene, exercise, or physical activity alone or in combination. Regarding the control groups, four studies included active control groups [23,31,32,48] and the rest, inactive control groups. Postural hygiene was the most common treatment, being present in all experimental groups. For the experimental group, the treatment was based exclusively on postural hygiene in fifteen studies [22,23,24,25,31,33,34,35,37,38,40,41,42,44,45], the combination of postural hygiene + exercise in six studies [30,32,39,46,47,49], the combination of postural hygiene + physical activity in one study [36], and the combination of postural hygiene + exercise + physical activity in two studies [43,48]. Regarding the types of postural hygiene, the most common were knowledge acquisition (theoretical and/or practical) in combination with postural habits training. All studies applied a group treatment.

As for the active control groups, the interventions were based on regular academic activities on related topics [31], postural hygiene [23], postural hygiene + exercise [32], and treatment as usual [48]. 

Concerning treatment time, in the experimental groups, the duration in weeks ranged from 1 week [22,47] to 96 weeks [35,36]. The intensity (hours/week of treatment) ranged from 0.16 [22,23] to 2.625 [48]. Total treatment time ranged from 0.16 hours [22,23] to 19 hours [32]. All experimental groups except one [40] had the number of sessions established before starting treatment, and all experimental groups except one [40] were based on homogeneous treatments to all participants. Only four studies included homework for participants to perform treatment at home [32,43,44,47]. 

With regard to external agents, in some studies external agents were involved in the treatment, such as teachers [23,24,30,31,35,36,37,40,41], family members [23], or both in the same intervention group [32,33]. In addition, the number of therapists who performed the treatment was diverse, ranging from one therapist [22,23,24,25,30,33,34,35,36,37,38,39,41,42,44,47,48] to six therapists [40].

### 3.3. Risk of Bias

All studies were judged to show some concerns except four studies with high risk of bias [22,43,45,48]. Three of the studies had high risk of bias in the first domain (randomization) [43,45,48]. One study obtained high risk of bias because it had some concerns in most of the domains [22]. All studies obtained some concerns in the second domain (deviations from intended interventions) except one that obtained high risk of bias [48]. One study obtained high risk of bias because it had some concerns in most of the domains [22]. All studies obtained low risk of bias in the fifth domain. For more information, see Table 2.

### 3.4. Effect Size

#### 3.4.1. Behavior

For the behavior variable, 19 studies were included, which resulted in 23 effect sizes. In the posttest, an overall effect size of d_+_ = 1.48 (95%CI: 0.40 to 2.56), with I^2^ = 98.94% of the total variability due to heterogeneity, was obtained. The 95% prediction interval ranged from −3.4 to 6.3. All studies except one yielded better results for the experimental group than for the control group [22]. One study obtained an extremely large effect size d_+_ = 13.03 [32]. Thirteen effect estimates revealed significant differences in favor of physiotherapy [24,30,32,34,35,41,42,43,45,46,47,48,49] (Figure 2). The Egger test was significant (*p* = 0.0090), with the asymmetry in the funnel plot suggesting potential publication bias (Figure 3). 

Given that the study by Méndez and Gómez 2001 [32] obtained an extreme effect size (d_+_ = 13.03), a sensitivity analysis excluding this study was performed. The new analysis resulted in a decrease in the effect size to d_+_ = 1.04 (95%CI: 0.64 to 1.43), with I^2^ = 91.54% of the total variability due to heterogeneity. The 95% prediction interval then ranged from −0.6 to 2.68. The Egger test for this analysis was not significant (*p* = 0.99).

Most studies that were included in meta-analysis 2 [15] reported statistically significant results in favor of physiotherapy, whereas studies included in meta-analysis 1 were more heterogeneous [14]. This may be because primary studies that were published after meta-analysis 1 considered the findings of that study. 

The two studies that included the most weeks of treatment (96 weeks) obtained better results in the experimental groups, with one reporting statistically significant results [35] while the other one did not [36]. The only study that obtained better results in the control group compared to the experimental group based its treatment on a single week [22]. Other studies that also applied single-week interventions obtained non-significant differences in favor of the experimental group [44] and another study did obtain significant differences in favor of the experimental group [46]. 

Regarding the number of hours/week of treatment, the study with the largest effect size was the study with the highest number of hours/week (2.375 h/week) [32]. The study with the highest number of hours/week of treatment obtained a large effect size with statistical significance in favor of the experimental group [48]. Regarding the studies with the lowest number of hours/week (0.16 h/week) [22,23], none obtained significant improvements and one study obtained better results for the control group [22].

Regarding the total number of hours, the study with the highest number of total hours of treatment (19 total hours) was the study with the largest effect size [32]. The second study with the highest number of total hours of treatment obtained statistical significance in favor of the experimental group [43]. In contrast, the two studies with the lowest total number of hours obtained non-significant results in favor of the control group [22] and non-significant results in favor of the experimental group [44].

The study with the largest effect size based its intervention on postural hygiene + exercise [32], the second study with the largest effect size only performed postural hygiene [24], and the third study with the largest effect size performed postural hygiene + exercise + physical activity [48]. Conversely, the three studies with the smallest effect sizes only performed postural hygiene [22,25,44].

#### 3.4.2. Moderator Analyses

##### Weighted ANOVA

Due to the high heterogeneity observed in the meta-analysis, a search for potentially moderating qualitative variables was conducted using ANOVA. Initially, the effects of different treatments (postural hygiene vs. postural hygiene + exercise) were assessed. Studies that incorporated postural hygiene in addition to exercise obtained better results than those that focused only on postural hygiene, although the difference between them was only marginally significant (*p* = 0.064). Furthermore, an evaluation of the postural hygiene methods (theory vs. theory + practical) was undertaken. Results suggest that studies implementing a combination of theoretical and practical approaches to postural hygiene achieved better outcomes compared to those relying solely on theoretical frameworks, although the difference between them was again only marginally significant (*p* = 0.086). Age groups (children vs. adolescents) were also analyzed through ANOVA, with no evidence of statistically significant differences between them (*p* = 0.60). Lastly, in the analysis exploring the impact of risk of bias (some concerns vs. high), there was no evidence of a difference between studies with “some concerns” and those assessed at high risk of bias (*p* = 0.53), however, the studies that received some concerns of risk of bias obtained statistically significant results, and the high risk of bias studies did not. Further information is available in Table 3.

##### Meta-Regression Analysis

Quantitative moderator variables were also analyzed using meta-regression. Specifically, the number of treatment weeks was examined, revealing no significant relationship with effect sizes (b_j_ = −0.0053, 95%CI −0.051 to 0.04, *p* = 0.810). Similarly, the number of hours per week of treatment (intensity) was not found to influence the effect sizes (b_j_ = 0.0735, 95%CI −0.277 to 1.747, *p* = 0.143). However, concerning the total number of treatment hours, we found evidence of a direct relationship (b_j_ = 0.395, 95%CI 0.209 to 0.581, *p* = 0.0003). Further details can be found in Table 4.

##### Knowledge

For the knowledge variable, 17 studies contributing 20 effect sizes were included. In the posttest, an overall effect size of d_+_ = 1.41 (95%CI: 1.05 to 1.76), with I^2^ = 86.03% of the total variability due to heterogeneity, was obtained. The 95% prediction interval ranged from −0.04 to 2.86. All studies obtained better results in the experimental group than in the control group. Twelve studies obtained significant results in the experimental group in comparison with the control group [22,25,30,31,32,33,37,38,41,46,48,49] (Figure 4). The studies with the longest effect size compared the physiotherapy group with an inactive control group [30,38,41,49]. These four studies were not the ones with the longest treatment duration, most hours of treatment per week, or highest total treatment hours compared to all others included. Of these, two included exclusively postural hygiene [30,38], while another two combined postural hygiene with exercise [41,49]. However, these four studies included both theory and practice in their treatment of postural hygiene. The Egger test was non-significant (*p* = 0.96), and the symmetry in the funnel plot shows the absence of publication bias (Figure 5). 

We observed that all studies that were included in the most recent meta-analysis 2 [15] reported statistically significant results in favor of physiotherapy, whereas the studies included in meta-analysis 1 were more heterogeneous [14]. As for the behavior variable, this may be because the studies that were published after meta-analysis 1 were influenced by its findings.

Conversely, within the subset of studies reporting non-significant results [24,35,36,39,40], two conducted the longest treatment periods, extending to 96 weeks [35,36] and 15 weeks of treatment [24]. However, in these five cases, details on intensity (hours/week) and total time of treatment were not provided. Notably, all of these studies mainly incorporated postural hygiene into their treatment strategies, with two exceptions. One study combined postural hygiene with physical activity [36], and another combined postural hygiene with exercise [39].

#### 3.4.3. Moderator Analyses

##### Weighted ANOVA

As a result of the high heterogeneity found in the meta-analysis, an analysis of the qualitative moderator variables by ANOVA was performed. First, the type of treatment (postural hygiene vs. postural hygiene + exercise) was analyzed, with no evidence of a difference between the options (*p* = 0.52). The type of postural hygiene (theoretical vs. theoretical + practical) was also analyzed, and again no evidence of a difference between categories was found (*p* = 0.96). In addition, the risk of bias categories were also analyzed (some concerns vs. high) without significant differences between them (*p* = 0.81). Last, an analysis of age groups (children vs. adolescents) was not feasible since all studies analyzed only included children. Further information is available in Table 5.

##### Meta-Regression Analysis

For quantitative moderator variables, the number of weeks of treatment was analyzed, resulting in a statistically significant result which surprisingly points to an inverse relationship (b_j_ = −0.013, 95%CI −0.022 to −0.003, *p* = 0.012). The number of hours per week of treatment was not found to influence the effect sizes (b_j_ = −0.083, 95%CI −0.355 to 0.189, *p* = 0.522). Last, total treatment time was also analyzed and was found not to influence the results (b_j_ = −0.003, 95%CI −0.091 to 0.098, *p* = 0.934). For more information, see Table 6.

### 3.5. Certainty of Evidence

The GRADE system was applied by two researchers independently with full agreement. Both variables, behavior and knowledge, obtained moderate certainty of evidence. For more information, see Appendix A.

## 4. Discussion

The aim of this meta-analysis was to quantify the effects of preventive physiotherapy on knowledge and behavior regarding back care in children and adolescents, for which 24 studies resulting in 28 reports were included.

Although behavior and knowledge have been studied previously via several meta-analyses [14,15], until this study not all studies carried out on this subject had been combined in a single meta-analysis, which is why this study has additional relevance. The results of this meta-analysis followed the same direction as the previously published meta-analyses, since this study obtained an overall effect size for the behavior variable of d_+_ = 1.48 (95%CI: 0.40 to 2.56) for the posttest, slightly higher than that of meta-analysis 1 (d_+_ = 1.33 (95%CI: 0.76 to 1.90)) and higher than that of meta-analysis 2 (d_+_ = 1.19 (95%CI: 0.62 to 1.76)). For the knowledge variable, in the present meta-analysis an overall effect size of d_+_ = 1.41 (95%CI: 1.05 to 1.76) was obtained, higher than in meta-analysis 1, which obtained an effect size of d_+_ = 1.29 (95%CI: 0.90 to 1.68), but lower than in meta-analysis 2, which obtained an effect size of d_+_ = 1.84 (95%CI: 0.58 to 3.09). It should be noted that meta-analysis 2 only included four studies in its analysis of knowledge, hence the effect size has such a wide range. In addition, the present study also provides the prediction interval, which was not present in the previous meta-analyses and included a more extensive number of studies in all the variables analyzed.

Whether the current study had a larger effect size than previous studies in general may be explained by the fact that the studies published since 2012 took into account the findings of meta-analysis 1, such as including more practical examples in their sessions and combining postural hygiene and exercise, instead of only postural hygiene. 

This study offered new insights into the behavior variable, for example, revealing that the studies that included postural hygiene in combination with exercise obtained better results, since in meta-analysis 1 the comparison was imprecise (only one study included postural hygiene in combination with exercise) and in meta-analysis 2 few studies were included and the result was not significant. Concerning the continuous moderating variables, meta-analysis 1 found that the number of weeks did not influence the effect size, intensity was close to statistical significance, and the total number of hours was included, and meta-analysis 2 also found that intensity was close to statistical significance, with the rest not being influential. However, the present study concluded that intensity is not significant, and neither is the number of weeks, although the total treatment time was.

Furthermore, the analysis of the moderator variables in the knowledge variable showed the same results in meta-analysis 1 and the present study that the combination of postural hygiene and exercise had better results than postural hygiene alone, but without significant differences. The present study showed more consistency in the different treatments (fourteen studies included postural hygiene and six studies included postural hygiene + exercise) compared to meta-analysis 1 (twelve studies included postural hygiene and three studies included postural hygiene + exercise). In addition, with respect to the type of teaching, meta-analysis 1 and the present one were the same, with a small difference in favor of theory only in comparison with theory + practical, with no significant differences between them. For quantitative variables, this meta-analysis discovered that the number of weeks negatively impacted effect size, a finding consistent with meta-analysis 1. Both studies agreed that neither treatment intensity nor total hours significantly affected knowledge effect size. Meta-analysis 2 did not perform moderator variable analysis due to the few studies included. 

A recently published systematic review included eight studies based on back care intervention in children and adolescents, examining the variables knowledge and behavior, among others [50]. The studies included in that review were also mostly included in the present study; those excluded did not have the statistical information needed for a meta-analysis or studied a different variable from those of interest in the present study. This systematic review defended that a back care education program was useful to improve children’s knowledge and behavior, highlighting the need to incorporate it in education programs, which is also in line with the present study. Another recent systematic review [13] argues that exercise is the most useful way to promote spinal health in the short term, but the combination of exercise with education is necessary for long-term management.

Considering the findings acquired in this study and combining them with those of the previous ones [14,15,50], we can develop strategies to improve knowledge and behavior regarding back care in this population. In this way, in order to improve both variables as a conjunct, it would be convenient to combine postural hygiene and exercise with theory and practice, reducing to a minimum the number of weeks of treatment and increasing the intensity and the total time of treatment. Although it is true that there are differences depending on whether we aim to improve knowledge or behavior separately, we recommend that when interventions are carried out to improve back care to prevent LBP, these two variables be addressed at the same time, since an improvement in knowledge can benefit behavior, mainly in the long term.

By using the GRADE approach to evaluate the certainty of the evidence, our study provided a better understanding of the effects of physiotherapy in back care and prevention of non-specific low back pain in children and adolescents. Since a moderate level of certainty of evidence was obtained, physiotherapists may be confident in the findings, being able to apply the insights provided in this study to their clinical practice. 

The main limitations of this study were the high heterogeneity found among the included studies, probably due to the variability among the studies (type and duration of treatment, etc.) as well as the fact that most of the studies obtained some concerns of risk of bias and a few had high risk of bias. Furthermore, the search strategy included studies up to 2020, as the aim of this study was to integrate the two previous meta-analyses to provide overarching conclusions in this field. Regarding the strengths, it is a study that encompasses all previously published studies, reaching a large sample size. In addition, two researchers separately performed the extraction of information, risk of bias analysis, and the application of the GRADE system. State-of-the-art statistical analysis techniques were used, and an exhaustive analysis of moderator variables was performed.

As far as we are aware, this is the most comprehensive meta-analysis to date, providing important insights that had not been discovered before. The results found in this study can serve physiotherapists, teachers, and policy makers to make precise interventions in the approach to back care and adapt their treatments to the study population.

## 5. Conclusions

Physiotherapy showed positive effects on behavior and knowledge related to back care to prevent NSLBP in children and adolescents. Postural hygiene and exercise were identified as the fundamental cornerstones of intervention. Treatments with the smallest number of weeks, the highest intensity, and the highest total number of treatment hours possible should be preferred.

## Figures and Tables

**Figure 1 healthcare-12-01036-f001:**
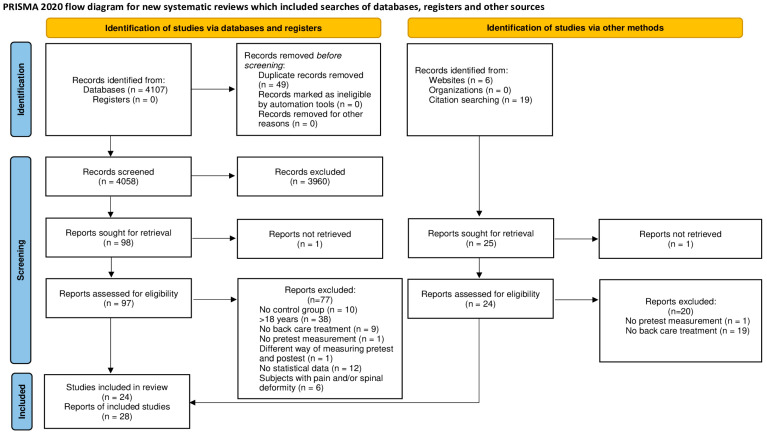
PRISMA flow diagram. Process of identification and selection of studies.

**Figure 2 healthcare-12-01036-f002:**
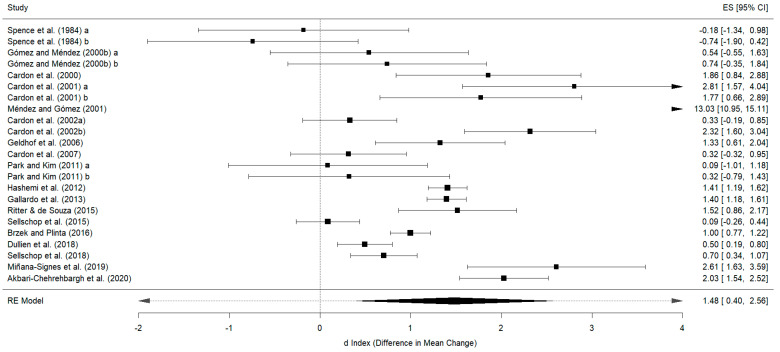
Forest plot of effect sizes for measures of behaviour in the posttest [22,23,24,25,30,32,33,34,35,36,41,42,43,44,45,46,47,48,49].

**Figure 3 healthcare-12-01036-f003:**
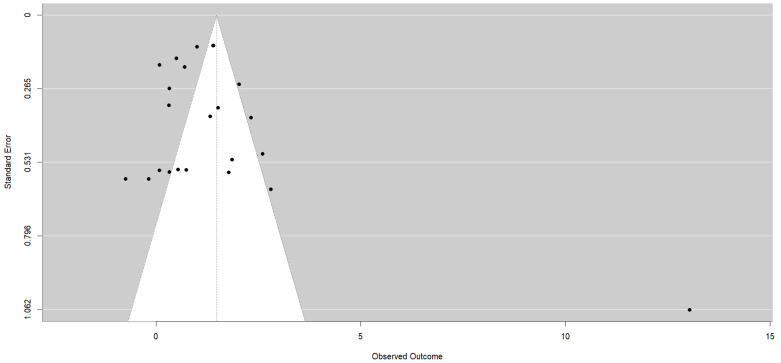
Funnel plot for the behaviour in the posttest.

**Figure 4 healthcare-12-01036-f004:**
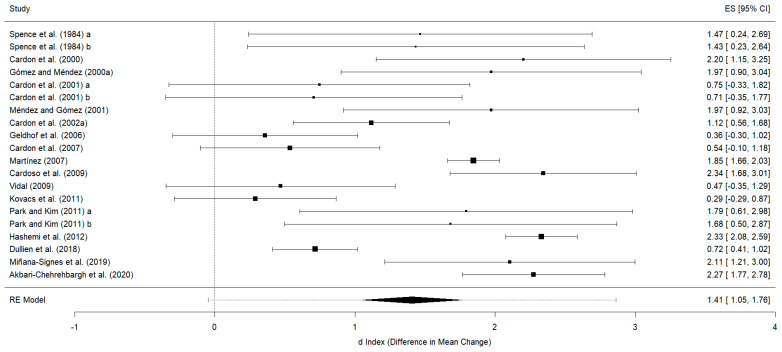
Forest plot of effect sizes for measures of knowledge in the posttest [22,24,25,30,31,32,33,35,36,37,38,39,40,41,46,48,49].

**Figure 5 healthcare-12-01036-f005:**
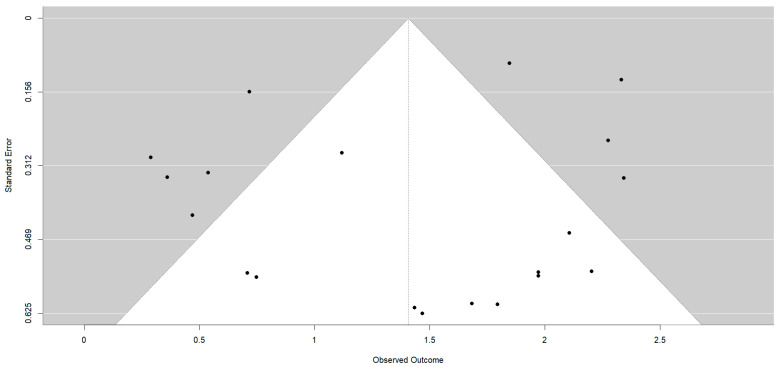
Funnel plot for the knowledge in the posttest.

**Table 2 healthcare-12-01036-t002:** Risk of bias of included studies.

Study	RoB1	RoB2	RoB3	RoB4	RoB5	RoBOverall
Spence et al., 1984 [22]	Some concerns	Some concerns	Some concerns	Some concerns	Low	High
Gómez and Méndez, 2000a [31]	Some concerns	Some concerns	Low	Low	Low	Some concerns
Gómez and Méndez, 2000b [23]	Some concerns	Some concerns	Low	Low	Low	Some concerns
Cardon et al., 2000 [30]	Some concerns	Some concerns	Low	Low	Low	Some concerns
Cardon et al., 2001 [24]	Some concerns	Some concerns	Low	Low	Low	Some concerns
Méndez and Gómez, 2001 [32]	Some concerns	Some concerns	Low	Low	Low	Some concerns
Cardon et al., 2002a [33]	Some concerns	Some concerns	Low	Low	Low	Some concerns
Cardon et al., 2002b [34]	Some concerns	Some concerns	Low	Low	Low	Some concerns
Geldhof et al., 2006 [35]	Some concerns	Some concerns	Low	Low	Low	Some concerns
Cardon et al., 2007 [36]	Some concerns	Some concerns	Low	Low	Low	Some concerns
Martínez, 2007 [37]	Low	Some concerns	Low	Low	Low	Some concerns
Cardoso et al., 2009 [38]	Some concerns	Some concerns	Low	Low	Low	Some concerns
Vidal, 2009 [39]	Low	Some concerns	Low	Some concerns	Low	Some concerns
Kovacs et al., 2011 [40]	Low	Some concerns	Low	Low	Low	Some concerns
Park and Kim, 2011 [25]	Low	Some concerns	Low	Some concerns	Low	Some concerns
Hashemi et al., 2012 [41]	Low	Some concerns	Low	Low	Low	Some concerns
Gallardo et al., 2013 [42]	Low	Some concerns	Low	Low	Low	Some concerns
Ritter & de Souza, 2015 [43]	High	Some concerns	Low	Some concerns	Low	High
Sellschop et al., 2015 [44]	Low	Some concerns	Low	Some concerns	Low	Some concerns
Brzek et al., 2016 [45]	High	Some concerns	Low	Some concerns	Low	High
Dullien et al., 2018 [46]	Low	Some concerns	Low	Low	Low	Some concerns
Sellschop et al., 2018 [47]	Low	Some concerns	Low	Some concerns	Low	Some concerns
Miñana-Signes et al., 2019 [48]	High	High	Low	Low	Low	High
Akbari-Chehrehbargh et al., 2020 [49]	Low	Some concerns	Low	Low	Low	Some concerns

**Table 3 healthcare-12-01036-t003:** Results of the weighted ANOVAs for the behavior, taking qualitative moderator variables as independent variables.

Variable	k	d_+_	95%CI		ANOVA Results
			LL	LU	
Type of treatment:					*F*(1, 21) = 3.80, *p* = 0.064
Postural hygiene Postural hygiene + exercise	16 7	0.848 2.914	−0.362 1.073	2.059 4.754	
Type of postural hygiene:					*F*(1, 21) = 3.23, *p* = 0.086
Theoretical Theoretical + practical	8 15	0.253 2.116	−1.490 0.850	1.996 3.381	
Age group:					*F*(1, 21) = 0.28, *p* = 0.60
Children Adolescents	20 3	1.596 0.765	0.409 −2.252	2.783 3.783	
Risk of bias:					*F*(1, 21) = 0.39, *p* = 0.537
High Some concerns	5 18	0.854 1.659	−1.508 0.414	3.216 2.904	

k = number of studies. d_+_ = mean coefficient alpha. LL and LU = lower and upper 95% confidence limits for d_+_. F = Knapp–Hartung’s statistic for testing the significance of the moderator variable.

**Table 4 healthcare-12-01036-t004:** Results of the simple meta-regressions for the behavior and magnitude measures, taking continuous moderator variables as predictors.

Predictor Variable	k	b_j_	CI.LL	CI.UL	F	p
Number of weeks of treatment	21	−0.0053	−0.051	0.040	0.059	0.81
Number of hours/week (intensity)	18	0.73	−0.277	1.747	2.370	0.14
Total time of treatment (magnitude)	19	0.395	0.209	0.581	20.09	0.0003

k = number of studies. b_j_ = regression coefficient of each predictor. CI.LL = confidence interval of lower limit. CI.UL = confidence interval of upper limit. F = Knapp–Hartung’s statistic for testing the significance of the predictor (the degrees of freedom for this statistic are 1 for the numerator and k − 2 for the denominator). p = probability level for the F statistic.

**Table 5 healthcare-12-01036-t005:** Results of the weighted ANOVAs for the knowledge, taking qualitative moderator variables as independent variables.

Variable	k	d_+_	95%CI		ANOVA Results
			LL	LU	
Type of treatment:					*F*(1, 18) = 0.416, *p* = 0.526
Postural hygiene Postural hygiene + exercise	14 6	1.334 1.574	0.901 0.924	1.767 2.22	
Type of postural hygiene:					*F*(1, 18) = 0.0022, *p* = 0.963
Theoretical Theoretical + practical	8 12	1.418 1.401	0.829 0.937	2.006 1.862	
Risk of bias:					*F*(1, 18) = 0.059, *p* = 0.810
High Some concerns	5 15	1.328 1.430	0.551 1.018	2.105 1.842	

k = number of studies. d_+_ = mean coefficient alpha. LL and LU = lower and upper 95% confidence limits for d_+_. F = Knapp–Hartung’s statistic for testing the significance of the moderator variable.

**Table 6 healthcare-12-01036-t006:** Results of the simple meta-regressions for the knowledge and magnitude measures, taking continuous moderator variables as predictors.

Predictor Variable	k	b_j_	CI.LL	CI.UL	F	p
Number of weeks of treatment	18	−0.013	−0.022	−0.0032	7.949	0.012
Number of hours/week (intensity)	15	−0.083	−0.355	0.189	0.433	0.52
Total time of treatment (magnitude)	16	0.0037	−0.091	0.098	0.0071	0.93

k = number of studies. b_j_ = regression coefficient of each predictor. CI.LL = confidence interval of lower limit. CI.UL = confidence interval of upper limit. F = Knapp–Hartung’s statistic for testing the significance of the predictor (the degrees of freedom for this statistic are 1 for the numerator and k − 2 for the denominator). p = probability level for the F statistic.

## Data Availability

The datasets generated during and/or analyzed during the current study are available from the corresponding author upon reasonable request.

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
