# Peer review of "Assessment of the Effects of Physiotherapy on Back Care and Prevention of Non-Specific Low Back Pain in Children and Adolescents: A Systematic Review and Meta-Analysis"

_healthcare, 2024, doi:10.3390/healthcare12101036_

Round 1
Reviewer 1 Report
Comments and Suggestions for Authors
I have reviewed the manuscript titled "Assessment of the effects of physiotherapy on back care and prevention of non-specific low back pain in children and adolescents: a systematic review and meta-analysis." This paper is notably interesting, particularly in its emphasis on the role of education and exercise in preventing this condition. I would suggest incorporating the following references to enrich the introduction and discussion sections:
- https://doi.org/10.1080/07853890.2022.2140453
- https://doi.org/10.1186/s12906-023-04061-1
In the materials and methods section, I noticed the absence of the Kappa index for assessing agreement among the authors who selected the articles. If feasible, it would be beneficial to include this index to enhance the section's rigor.
Additionally, although a bias analysis has been conducted, the methodological quality of the included studies appears unassessed. I recommend employing the PeDRo scale for evaluating methodological quality, as this is a well-regarded tool in the field and is also one of the databases you utilized.
Regarding the meta-analysis, consider adding analyses of subgroups (such as exercise alone, education alone, both, and other interventions) to address qualitative heterogeneity. Ensure that the existing analyses are retained, and supplement them with the new ones to broaden the study's scope.
The discussion and conclusions are adequately drafted; however, there is room to deepen the exploration of risk factors, compare effects across different age groups, and discuss these elements more thoroughly.
Upon implementation of these revisions, I am willing to review the manuscript again with the intention of moving towards acceptance for publication.
Author Response
Question 1. I have reviewed the manuscript titled "Assessment of the effects of physiotherapy on back care and prevention of non-specific low back pain in children and adolescents: a systematic review and meta-analysis." This paper is notably interesting, particularly in its emphasis on the role of education and exercise in preventing this condition. I would suggest incorporating the following references to enrich the introduction and discussion sections:
- https://doi.org/10.1080/07853890.2022.2140453
- https://doi.org/10.1186/s12906-023-04061-1
Response: we have taken this comment into account and both studies have been included in the introduction section (references 11 and 12).
Question 2: In the materials and methods section, I noticed the absence of the Kappa index for assessing agreement among the authors who selected the articles. If feasible, it would be beneficial to include this index to enhance the section's rigor.
Response: We have reported the value of the Kappa index in the paper.
Question 3. Additionally, although a bias analysis has been conducted, the methodological quality of the included studies appears unassessed. I recommend employing the PEDro scale for evaluating methodological quality, as this is a well-regarded tool in the field and is also one of the databases you utilized.
Response: methodological quality and risk of bias are used as interchangeable terms in the systematic review arena. We agree that the PEDro scale is a useful tool for assessing this construct, but we note that with the information from the Cochrane tool applied in our review the reader already has adequate information to determine the quality of each of the included studies.
Question 4. Regarding the meta-analysis, consider adding analyses of subgroups (such as exercise alone, education alone, both, and other interventions) to address qualitative heterogeneity. Ensure that the existing analyses are retained, and supplement them with the new ones to broaden the study's scope.
Response: We agree that a subgroup analysis according to the type of treatment provides very relevant information, especially for clinical practice. In table 3 (behavior variable) and table 5 (knowledge variable), the type of treatment was analyzed in two groups: postural hygiene and postural hygiene + exercise. Only these two groups have been analyzed since no study provided exercise alone, and the most frequently reported intervention was postural hygiene alone or the combination of postural hygiene + exercise. It would be interesting in the future to carry out studies with a greater diversity of treatments, in order to be able to make more precise comparisons.
Question 5. The discussion and conclusions are adequately drafted; however, there is room to deepen the exploration of risk factors, compare effects across different age groups, and discuss these elements more thoroughly.
Response: We agree that age is an important factor, so separating into children and adolescents for the analyses is necessary. In Table 3 (behavior), an analysis of comparison between groups can be seen, being 20 studies that included only children and 3 only adolescents. We would have liked to perform this analysis also in Table 5 (Knowledge), but this was not done since all the studies included only children; this information can be seen in the results section 3.4.3.
Regarding risk factors, we agree that they are essential elements to be studied, especially when we talk about the youth population, however, and since this is not the main objective of the study, we have not explored this area in depth, although it would be interesting in the future to carry out a study focusing only on risk factors.
Reviewer 2 Report
Comments and Suggestions for Authors
I believe that this is an important finding to provide the effects of preventive physiotherapy interventions on improving behavior and knowledge related to back care and prevention of NSLBP in children and adolescents. However, your manuscript has many serious problems. Analyzes should be performed according to rigorous methods.
#1
Although the authors declared compliance with PRISMA 2020, certainty of evidence results were missing. GRADE is generally used for certainty of evidence. Results should be reported for risk of bias, imprecision, indirectness, and publication bias, as well as heterogeneity (inconsistency). Additionally, the degree of certainty should be discussed. I cannot trust your work without certainty of evidence.
#2
Selection criteria for 3arm clinical trials are not described. Are these included in the meta-analysis? Please describe exactly how you conducted your analysis. I recommend performing a network meta-analysis or removing these clinical trials from the analysis.
Author Response
I believe that this is an important finding to provide the effects of preventive physiotherapy interventions on improving behavior and knowledge related to back care and prevention of NSLBP in children and adolescents. However, your manuscript has many serious problems. Analyzes should be performed according to rigorous methods.
Question 1. Although the authors declared compliance with PRISMA 2020, certainty of evidence results were missing. GRADE is generally used for certainty of evidence. Results should be reported for risk of bias, imprecision, indirectness, and publication bias, as well as heterogeneity (inconsistency). Additionally, the degree of certainty should be discussed. I cannot trust your work without certainty of evidence.
Response: We agree that the GRADE system of certainty of evidence provides valuable information and improves the quality of the study. For this reason, it has been applied and added to the methods, results and discussion sections and to the supplementary files (Table S4).
Question 2. Selection criteria for 3arm clinical trials are not described. Are these included in the meta-analysis? Please describe exactly how you conducted your analysis. I recommend performing a network meta-analysis or removing these clinical trials from the analysis.
Response: Four included studies were 3-arm trials with two experimental groups. For these, the effect size was calculated separately for each experimental group in relation to the control group. While this violates the independence assumption required when applying standard meta-analytic techniques (as each participant allocated to control groups in those studies contributed to two effect sizes), we note that the extent of multiplicity in our database – and consequently the potential for statistical dependency – is minimal given the reduced number of studies affected, and the small size of the groups involved compared to the rest of included studies (see Table 1). Therefore, standard meta-analytic models were preferred for this review over other methodological options that either involve loss of information or additional statistical complexity.
This explanation has been included in the methods section (2.6).

Round 2
Reviewer 1 Report
Comments and Suggestions for Authors
The authors have correctly incorporated all the comments.
Congratulations on your work.
Reviewer 2 Report
Comments and Suggestions for Authors
The author has resolved the relevant issues.